# Inhibition of 37/67kDa Laminin-1 Receptor Restores APP Maturation and Reduces Amyloid-β in Human Skin Fibroblasts from Familial Alzheimer’s Disease

**DOI:** 10.3390/jpm10040232

**Published:** 2020-11-16

**Authors:** Antaripa Bhattacharya, Antonella Izzo, Nunzia Mollo, Filomena Napolitano, Adriana Limone, Francesca Margheri, Alessandra Mocali, Giuseppina Minopoli, Alessandra Lo Bianco, Federica Di Maggio, Valeria D’Argenio, Nunzia Montuori, Antonio Lavecchia, Daniela Sarnataro

**Affiliations:** 1Department of Molecular Medicine and Medical Biotechnology, University of Naples “Federico II”, Via S. Pansini 5, 80131 Naples, Italy; antaripa1210@gmail.com (A.B.); antonella.izzo@unina.it (A.I.); nunzia.mollo@unina.it (N.M.); adriana.limone1996@gmail.com (A.L.); giuseppina.minopoli@unina.it (G.M.); dimaggio@ceinge.unina.it (F.D.M.); 2Department of Translational Medical Sciences, University of Naples “Federico II”, Via S. Pansini 5, 80131 Naples, Italy; filomena-napolitano88@hotmail.it (F.N.); nmontuor@unina.it (N.M.); 3Department of Experimental and Clinical Biomedical Sciences, University of Florence, 50134 Florence, Italy; francesca.margheri@unifi.it (F.M.); alessandra.mocali@unifi.it (A.M.); 4Department of Pharmacy, “Drug Discovery Lab”, University of Naples “Federico II”, Via D. Montesano 49, 80131 Naples, Italy; alessandra.lobianco8@gmail.com (A.L.B.); antonio.lavecchia@unina.it (A.L.); 5CEINGE-Biotecnologie Avanzate Scarl, Via G. Salvatore 486, 80145 Naples, Italy; dargenio@ceinge.unina.it; 6Department of Human Sciences and Quality of Life Promotion, San Raffaele Open University, Via di Val Cannuta 247, 00166 Rome, Italy

**Keywords:** Alzheimer’s disease, amyloid-β, amyloid precursor protein APP, NSC47924, 37/67kDa laminin-1 receptor inhibitor

## Abstract

Alzheimer’s disease (AD) is a fatal neurodegenerative disorder caused by protein misfolding and aggregation, affecting brain function and causing dementia. Amyloid beta (Aβ), a peptide deriving from amyloid precursor protein (APP) cleavage by-and γ-secretases, is considered a pathological hallmark of AD. Our previous study, together with several lines of evidence, identified a strict link between APP, Aβ and 37/67kDa laminin receptor (LR), finding the possibility to regulate intracellular APP localization and maturation through modulation of the receptor. Here, we report that in fibroblasts from familial AD (fAD), APP was prevalently expressed as an immature isoform and accumulated preferentially in the transferrin-positive recycling compartment rather than in the Golgi apparatus. Moreover, besides the altered mitochondrial network exhibited by fAD patient cells, the levels of pAkt and pGSK3 were reduced in respect to healthy control fibroblasts and were accompanied by an increased amount of secreted Aβ in conditioned medium from cell cultures. Interestingly, these features were reversed by inhibition of 37/67kDa LR by NSC47924 a small molecule that was able to rescue the “typical” APP localization in the Golgi apparatus, with consequences on the Aβ level and mitochondrial network. Altogether, these findings suggest that 37/67kDa LR modulation may represent a useful tool to control APP trafficking and Aβ levels with implications in Alzheimer’s disease.

## 1. Introduction

Extracellular amyloid plaques formed by deposits of Aβ peptide and intracellular neurofibrillary tangles, composed of hyperphosphorylated tau protein, represent the major neuropathologic event characterizing Alzheimer’s disease (AD) [1].

Aβ derives from a sequential proteolytic cleavage of amyloid precursor protein (APP) by β- and γ-secretases. Mutations in Aβ, as well as in APP, near the β- and γ-secretase sites [2], together with duplication of the *APP* locus, give rise to AD [3]. The overwhelming majority of dominant mutations causing familial AD occurs in three genes: *APP*, *PSEN1* (*Presenilin-1*), and *PSEN2* (*Presenilin-2*). Interestingly, mutations in PSEN-1 and -2 have been described to increase the Aβ_42_/Aβ_40_ ratio [4,5].

To date, 19 different AD causing mutations in the γ-secretase enzyme PSEN-2 have been reported and were shown to alter the APP processing by increasing both the total level of Aβ and the Aβ_42/40_ ratio [6,7], suggesting that the dysregulation of Aβ production from APP can be considered a common effect of these mutations.

In neurons, defects in APP trafficking and processing leading to Aβ production are likely the most common cause of AD; thus, understanding how APP is targeted to a defined cellular compartment and identifying the mechanisms that control the Aβ generation, could be a key for new therapies [8,9].

We have demonstrated that APP posttranslational modifications, strictly related to subcellular localization, and intracellular trafficking are extremely important for correct protein maturation and processing [10,11,12,13,14,15]. In this context, the already acknowledged role of 37/67kDa laminin receptor (LR) in the internalization and cytotoxicity of Aβ [16,17], together with our recent finding of its interaction with APP and its involvement in APP localization and maturation in neuronal cells [10], prompted us to analyze the effects of a specific inhibitor of 37/67kDa LR on the localization and processing of APP, as well as on Aβ generation, in fibroblast cell lines from familial AD patients compared with healthy unaffected ones.

We first characterized the absence of any pathogenic variant in *APP, PSEN1,* and *PSEN2* genes in one of the two cell lines from familial AD (here named fAD1). Then, we used another fibroblast cell line carrying the *PSEN2* M239V pathogenic mutation for familial AD (here named fAD3). Contrary to control fibroblasts (unAD, unaffected) where APP was localized in the Golgi apparatus, we found that in fAD fibroblast cell lines, APP lost its Golgi localization resulting mainly distributed in transferrin-positive recycling endosomes. The use of a specific inhibitor of 37/67kDa LR, NSC47924, completely rescued the localization of APP in the Golgi complex and restored the APP maturation, which was partially lost in fAD fibroblasts. In addition, the inhibitor was able to improve the mitochondrial network organization in terms of volume and number, to almost the same level of healthy individuals. Finally, the treatment with NSC47924 drastically reduced secreted Aβ levels in fAD fibroblasts culture media and inactivated Akt signaling with reduction of Ser9-pGSK3β, revealing the receptor a promising target for AD.

## 2. Results

### 2.1. Sanger Sequencing for APP, PSEN1 and PSEN2 in fAD1 Fibroblast Cell Line

To assess the presence of any pathogenic variant associated with AD onset, *APP*, *PSEN1,* and *PSEN2* genes’ sequences were analyzed in a fAD1 fibroblast cell line (Methods). As shown in Table 1, two variants were found in *APP*, two in *PSEN1,* and nine in *PSEN2*.

All *APP* and *PSEN1* variants were in the non-coding regions; they were all classified as benign according to ACMG criteria. In *PSEN2*, we found 9 variants, of which 6 were intronic and 3 were exonic; the latter were all synonymous variants with no alteration at the protein level. No pathogenic variants were found in all 3 analyzed genes.

Recent studies report that causative gene mutations associated with familial AD, have been identified in *APP, PSEN1,* and *PSEN2*. However, mutations in these genes explain just a small percentage of all fAD cases. This finding suggests the existence of other, inherited, disease-predisposing genes [18]. Since we did not find any mutation in the main candidate genes (*APP, PSENs*), further experimentations will be needed to search for gene/s possibly involved in the fAD1 phenotype.

### 2.2. Mature/Immature APP Isoform Ratio is Reduced in fAD Fibroblasts and NSC47924 Restores APP Isoform Ratio to that of unAD Fibroblasts

Since APP trafficking through the secretory pathway is a warranty for proper protein maturation, which in turn is crucial for protein processing [19], to analyze APP expression level and posttranslational modifications, we employed human skin fibroblasts from familial fAD1 and fAD3 individuals (see Materials and Methods for a detailed description).

As expected from the analysis of SDS-PAGE gel electrophoresis [10], the APP migrated as two different bands corresponding to immature (~100 kDa) and mature glycosylated isoform (~130 kDa) (Figure 1).

Densitometric analysis of bands obtained by western blotting procedure with anti-APP antibody was performed to quantify the ratio between APP isoforms. The percentage of mature APP, with respect to total bands, is reported in the lower panel of Figure 1.

By using tubulin as a loading control, we found that the mature/immature APP ratio was significantly lower in both fAD fibroblast cell lines (0.26 ± 0.04 in fAD1 and 0.20 ± 0.03 in fAD3) with respect to the unAD controls (0.42 ± 0.08, *p* < 0.05), indicating that the APP was not able to completely mature along the secretory pathway, where normally it should traffic [20]. Incubation of cells with NSC47924 inhibitor, significantly rescued the APP ratio values (0.55 ± 0.2 in fAD1 and 0.45 ± 0.08 in fAD3, versus 0.45 ± 0.1 in unAD) (Table 2), strongly suggesting the molecule is working possibly by “correcting” the trafficking of the APP in mutant cells.

The fact that the inhibitor effects were elicited both in fAD1, where no pathogenic *APP* or *PSEN* mutations were found, and in fAD carrying pathogenic *PSEN2* mutation, indicates that localization of the APP was independent of the expression of wild-type or mutated PSENs.

### 2.3. APP Is Mainly Localized in Recycling Endosomes Rather than in the Golgi Apparatus in fAD Fibroblasts

Our finding that APP was not completely mature in fAD cells, led us to speculate that APP was not able to be modified in the Golgi apparatus. Thus, to verify this hypothesis, we analyzed the intracellular distribution of APP in mutant cells, by employing fluorescence microscopy using different markers of the intracellular organelles. In agreement with previous observations in neuronal and non-neuronal cells [10,20,21], in control fibroblasts from unaffected donors, we found a significant colocalization of APP with Giantin, a marker of the Golgi complex (Pearson Correlation Coefficient, PCC = 0.86) (Figure 2).

On the contrary, in patient fibroblasts (both fAD1, Figure 3 and fAD2/3, Appendix A) APP lost its Golgi distribution to relocate into recycling endosomes (PCC = 0.92 for APP/Tfr), which were positive for Transferrin, a typical marker of recycling endosomes (Figure 3; Appendix A, upper panels, untreated).

Because we have previously described a link between APP subcellular localization/maturation and the 37/67kDa laminin receptor in neuronal cells [10], we tested the effects of a specific inhibitor of the receptor in fAD cells compared to unAD fibroblasts.

Interestingly, the use of NSC47924 inhibitor [22] restored the localization of APP in fAD fibroblasts relocating APP to the Golgi complex (Figure 2 and Figure 3; Appendix A, bottom panels, NSC47924 treated, PCC = 0.88 for APP/Giantin). Consequently, after treatment with the inhibitor, the APP lost the colocalization with the recycling endosomes (Figure 2 and Figure 3; Appendix A, bottom panels, PCC = 0.29 for APP/Tfr), which are critical intracellular compartments for peptide metabolism and protein function [23]. Thus, these results indicate that in AD fibroblasts, the “typical” Golgi APP intracellular localization is restored by a small molecule which is a specific inhibitor of the 37/67kDa LR.

### 2.4. The Mitochondrial Network Is Compromised in fAD Fibroblasts and Improved by Inhibition of 37/67kDa LR

Because mitochondrial morphology and number [24], as well as mitochondrial biogenesis and transport along the neuronal axon [25], were found to be severely compromised in AD [26], as well as in aging and other neurodegenerative diseases [27,28], we analyzed the mitochondrial network organization in fAD fibroblasts. Because the mitochondrial function is linked to the organization of the mitochondrial network [29], we examined the mitochondrial morphology by using the Mitotracker Red CMXRos probe (Figure 4).

The morphology of the mitochondrial network was profoundly altered in both fAD fibroblasts compared to unAD cells (Figure 4A). Three-dimensional (3D) reconstructions in the patient cells revealed that the mitochondrial network exhibited thinner and shorter tubules with respect to unAD cells, losing the typical interconnected tubular structure (Figure 4A). These data were strengthened by measuring the volume and number of mitochondria (Figure 4B,C), which were reduced in fAD compared to unAD, and then restored by the 37/67kDa LR inhibitor NSC47924 (Figure 4A, right panel, and Figure 4B,C).

### 2.5. Inhibitor Treatment Inactivates Akt Signaling with Consequent Activation of GSK3β Pathway

The mitochondrial pool of GSK3β has been involved in mitochondrial functions [30] and seems to be central in AD pathogenesis through changes in its phosphorylation state, as well as expression levels [31]. Starting from these observations, we investigated the phosphorylation status of GSK3β in Serine 9, which is indicative of GSK inactivation. In agreement with previous observation in neuronal cells [32], we found that pGSK3β levels were lower in fAD fibroblasts compared to unAD ones and that NSC47924 treatment decreased pGSK3β with respect to untreated samples (Figure 5A).

Furthermore, the analysis of Akt activation by western blot under NSC47924 treatment, revealed that pAkt was reduced with respect to untreated control conditions (Figure 5B).

### 2.6. The Amount of Secreted Aβ Is Significantly Higher in fAD Cells and Is Reduced by NSC47924 Treatment

The detection of Amyloid-β peptide and the identification of molecules capable to interfere with its generation, represent an important approach to study AD.

Herein, we analyzed by ELISA assays the levels of Aβ in conditioned media obtained from cultured skin fibroblasts of fAD patients and healthy controls (Figure 6). This analysis was conducted by ELISA assay because this method is able to capture Aβ species closer to their native conformation in vivo.

The assay relies on the use of an anti-Aβ monoclonal antibody for capture. To increase the specificity of the ELISA, we utilized non-immune IgG as a control, and in two independent assays, two different antibodies specifically recognizing Aβ peptide (6E10 in Figure 6 and 4G8 in Appendix A). In parallel, anti-Aβ antibody binding to BSA-coated wells was also evaluated and the absorbance readings subtracted, as control for binding specificity. Figure 6 shows a significant percentage increase of anti-Aβ antibody absorbance value over non-immune control IgG in fAD conditioned medium, with respect to healthy controls.

Next, we analyzed the effects of NSC47924 inhibitor on Aβ levels in conditioned medium from healthy and fAD fibroblasts. The treatment with NSC47924 led to a reduction of the expression levels of Aβ in both healthy control and fAD cells (* *p* ˂ 0.05).

Furthermore, besides the relocation of APP in the Golgi apparatus, we found that inhibitor treatment did not induce any accumulation of amyloid peptide inside fAD cells (Appendix A, bottom right panel, APP-6E10/Giantin).

## 3. Discussion

In this study, we investigated the effects of a specific inhibitor of 37/67kDa LR [33] on APP maturation and Aβ generation in fibroblasts from patients affected by familial Alzheimer’s disease (carrying *PSEN2* M239V pathogenic mutation, fAD2, fAD3 or not fAD1), compared with healthy unaffected controls.

Fibroblasts have been employed as an in vitro model for neurological diseases [34], and particularly for AD [35,36,37].

Since fibroblasts are the most commonly used primary somatic cell type for the generation of induced pluripotent stem cells, we hope to obtain a better characterization of biological processes related to NSC47924 action on fAD patient cell lines, to plan subsequently, reprogram and differentiate patient fibroblasts. Comparison of the effects of different LR inhibitor molecules on different cell types that may be considered to be non-relevant for the disease, such as fibroblasts, or more relevant to the disease, such as neurons differentiated from iPSCs, will allow us to develop more predictive in vitro systems for drug discovery. Our results obtained in fibroblasts represent the first step to reinforce the value of utilizing human iPSCs in drug discovery to improve translational predictability.

Previous studies have already reported a critical role for 37/67kDa LR in APP processing through direct interaction with the γ-secretases and indirect interaction with the β-secretase [38]. The role of 37/67kDa LR highlights the importance of identifying therapeutics that target the receptor [39], which has also been previously shown by us to be an interactor and regulator of prion protein trafficking [22,40,41]. In addition, our previous data showing the interaction between 37/67kDa LR and APP in mouse neuronal cells [10], together with the finding of regulation of APP maturation through receptor inhibition, prompted us to analyze the role of 37/67kDa LR in AD by challenging the specific receptor inhibitor NSC47924 (previously found active on prion protein trafficking) [22], with cells derived from fAD patients.

Protein maturation through the secretory pathway is strictly related to its intracellular trafficking. Since APP maturation is crucial for correct protein processing and coordination between the non-amyloidogenic and amyloidogenic pathway, we decided to test the effects of two LR inhibitors in pathological conditions, human cells derived from AD affected individuals.

After pilot experiments on APP using two different and previously selected analogs of LR inhibitors in mouse neuronal cells (NSC48478 and NSC47924) [10,22,33], only NSC47924 has been found to exert effects on trafficking and maturation of APP in human cells, restoring the “correct” ones that were lost in affected cells. Therefore, in the present study we decided to report only the active molecule NSC47924.

According to previous findings reporting the expression of APP as two major isoforms, corresponding to immature unglycosylated and mature modified isoform [10,42], we found APP migrating as two bands on SDS-PAGE. Differently from this observation, APP was expressed prevalently as an immature isoform in AD affected fibroblasts, suggesting an alternative trafficking pathway for APP in fAD cells with respect to unaffected cells.

Moreover, in line with other reports, where in physiological conditions, APP was localized in the Golgi apparatus [10,20,21,43], we found a prevalent distribution of APP in the Golgi of unAD cells. On the contrary, the APP is mainly colocalized with Tfr-positive recycling endosomes in the fAD fibroblast cell lines analyzed, suggesting a different trafficking pathway in disease conditions. This finding is extremely important if we consider previously reported data from Das et al., describing the convergence between the APP and the β-secretase BACE1 in recycling endosomes of neurons from brains with AD [44].

Interestingly, inhibition of 37/67kDa LR by NSC47924 was able to reverse the APP isoform ratio, restoring the levels almost to that of unAD cells.

APP modifications and trafficking are mutually regulated, contributing in turn to the modulation of Aβ generation [45]. The acquisition of APP post-translational modifications occurs mainly during the passage through the secretory pathway in neurons [42,46], where the protein can undergo glycosylation which starts in the ER to proceed through the Golgi, where sulfation and phosphorylation occur. Since we found that a reduced level of mature APP in AD-affected cells was accompanied by a mislocalization of the APP from the Golgi and consequent accumulation into recycling endosomes (Figure 3, Appendix A), we speculated that the APP was not able to be correctly modified for maturation probably because of its inability to transit through the secretory pathway and to reach the Golgi apparatus, where the majority of the APP modifications occurs.

Therefore, the possibility to control APP trafficking (by inhibiting the receptor LR, as well as we have previously done in neuronal cells) [10], would be reflected in APP maturation changing. Our hypothesis has been confirmed by showing that after inhibitor treatment, the APP recovered its Golgi localization and its molecular weight at the same level of unaffected cells (Figure 2 and Figure 3).

Processing of the APP by α-, β- and γ-secretases generating APP-CTFs (C-terminal fragments) and Aβ, is strictly related to the subcellular localization and trafficking of APP. Because the secretases have different distribution within the cells, the itinerary followed by the APP is crucial for determining the amount of APP fragments. Furthermore, CTFs have been found localized to subcellular microdomains between the ER and mitochondria [47], and they can lead, as well as Aβ, to mitochondrial dysfunction and morphology alteration in in vivo and in vitro AD study models [48,49,50]. According to findings in the literature, we observed an altered mitochondrial network in fAD cells. Therefore, we speculated that if in AD-affected cells we would be able (by using LR inhibitor) to reverse the subcellular localization of the APP and, consequently, its trafficking to that of healthy cells, we should have observed a restoring of cellular phenotype resembling one of healthy conditions, including mitochondrial network phenotype.

Given the key role of mitochondria in protein misfolding diseases [51,52], by means of immunofluorescence analysis on imaged fibroblasts, we have evaluated the mitochondrial network in terms of number and volume of mitochondria. In fAD cells, we observed a significant decrease of 3D objects volume and number, indicating mitochondrial dysfunction in disease conditions. Again, 37/67kDa LR inhibition almost completely rescued mitochondrial values to that of control fibroblasts by increasing both mitochondrial volume and number. This feature was accompanied by the finding of Akt inactivation and consequent pGSK3β decrease in fAD cells. Although several lines of evidence report increased activity of this enzyme in AD [32,53,54,55], we should acknowledge that direct evidence for this is still limited, since some studies find no change in GSK3 activity [56] or reduced GSK3 activity in AD [57].

Moreover, the ability of NSC47924 to “correct” the localization and maturation of APP and the consequent inactivation of the Akt pathway and regulation of GSK3β, can be critical for APP processing and reveal NSC47924 as a useful compound to be tested for Aβ production. Indeed, we challenged the inhibitor with fAD cells to test Aβ levels in cell culture media in comparison with unAD. Besides the fact that, as expected and previously found [58,59], the Aβ levels were higher in AD with respect to healthy conditions, the use of NSC47924 induced a significant decrease of Aβ levels in fAD. Furthermore, our finding that 37/67kDa LR inhibition did not induce any accumulation of amyloid peptide inside fAD cells, suggests that the reduction of Aβ in the cell culture media was likely due to impairment of Aβ production rather than to a lack of its secretion.

However, although further studies will be needed to validate 37/67kDa LR as a potential target for Alzheimer’s disease, since NSC47924 is essentially intended to act on CNS, it will be interesting to challenge the inhibitor with in vivo conditions, where the cellular environment governs the pathophysiology of the disease.

## 4. Materials and Methods

### 4.1. Sanger Sequencing

The coding regions and the boundary intronic regions of *APP*, *PSEN1* and *PSEN2* genes we sequenced by direct Sanger sequencing, and 17, 11, and 10 primer pairs were designed to amplify the above-mentioned regions of *APP*, *PSEN1,* and *PSEN2,* respectively (Appendix A). Genomic DNA was obtained by a pellet of at least 5 × 10^5^ cells using the MasterPure Complete DNA purification kit (Epicentre Biotechnologies, Madison, WI, USA) following the manufacturer’s instructions.

Each amplicon was individually amplified, assessed for quality on 2% agarose gel, and purified before sequencing reactions. Direct sequencing was performed with the ABI 3100 capillary sequencer (Applied Biosystems Inc., Foster City, CA, USA), and sequence data analysis was carried out using the SeqMan software (DNASTAR, Inc., Madison, WI, USA). Next, DNA variants were categorized accordingly to Ensembl (https://www.ensembl.org/index.html), ClinVar (https://www.ncbi.nlm.nih.gov/clinvar/) and dbSNP (https://www.ncbi.nlm.nih.gov/SNP/) databases and the possible impact of specific variations at the protein level was predicted using VarSome (https://varsome.com) and or Human Splicing Finder (http://www.umd.be/HSF/HSF.shtml) tools.

A full list of the primer pairs used to amplify *APP*, *PSEN1* and *PSEN2* genes are illustrated in Appendix A.

### 4.2. Reagents and Antibodies

Cell culture reagents were purchased from Gibco Laboratories (Grand Island, NY, USA). Transferrin Alexa-594- conjugated (Tfr), cy2-, cy5-conjugated secondary Abs and Mitotracker Red CMXRos were from Invitrogen (Molecular Probes, Eugene, OR, USA). The anti-Giantin antibody was from StressGen Biotechnologies Corp (Victoria, BC, Canada). Anti-β-tubulin antibody was from Abcam. Anti-GSK, anti-pGSK3β Ser9 and DAPI were from Cell Signalling Technology. Anti-APP rabbit polyclonal antibody A8717 was from Sigma-Aldrich (St. Louis, MO, USA). Anti-Aβ antibody 6E10 (previously Covance, SIG-39320) and 4G8 were from Biolegend. All other reagents were from Sigma Chemical Co. (St. Louis, MO, USA). NSC47924 [1-((4-methoxyanilino)methyl)-2-naphtol] was obtained from the NCI/DTP Open Chemical repository (http://dtp.cancer.gov), dissolved in DMSO and stored at −20 °C.

### 4.3. Cell Culture and Drug Treatment

Fibroblasts were derived directly from the skin punch biopsy of the Italian patients carrying (fAD2: male with onset AD 48 years, and fAD3: female with onset AD 45 years) or not (fAD1, female, with unknown onset AD) the *PSEN2* M239V mutation. Written informed consent was obtained from each patient. All fibroblasts were derived from patients with a clinical diagnosis of probable early-onset fAD according to the criteria established by the Diagnostic and Statistical Manual of Mental Disorders (4th edition, DSM IV) [American Psychiatric Association, *Diagnostic and Statistical Manual of Mental Disorders*, American Psychiatric Association, Washington, DC, USA, 4th edition, 1994], the National Institute of Neurological and Communicative Disorders and Stroke, and reevaluated according to the NIA-Alzheimer’s Association workgroups on diagnostic guidelines for AD [1]. The cells were grown in Dulbecco’s modified Eagle’s medium (DMEM), with 4500 mg/glucose/L, 110 mg sodium pyruvate and L-glutamine (SIGMA D6429), supplemented with 10% fetal bovine serum. For inhibitor NSC47924 treatment, the cells were washed in serum free medium, incubated for 30 min at room temperature in Areal medium (13.5 g/L of Dulbecco’s modified eagle’s medium with glutamine SIGMA-D-7777 without NaHCO_3_, 0.2% BSA and 20 mM HEPES, final pH 7.5) and for further 24 h at 37 °C under 5% CO_2_ in the presence of 20 μM inhibitor in DMEM supplemented with 1% serum. As control cells, fibroblasts from 2 unaffected individuals (unAD) were derived and grown in the same conditions. All the experiments were performed in 3 lines of fAD and 2 of unAD.

### 4.4. Indirect Immunofluorescence and Confocal Microscopy

Fibroblasts were cultured to 50% confluence in a growth medium, washed in PBS, fixed in 4% paraformaldehyde (PFA), permeabilized or not with 0.1% TX-100 for 30 min (where indicated) and processed for indirect immunofluorescence using specific antibodies 30 min in PBS/BSA 0.1%. The cells were incubated with rabbit anti-APP (A8718) antibody and markers of intracellular organelles, followed by incubation with fluorophore-conjugated secondary antibodies. When Tfr-Alexa594 was incubated in cell culture media, APP (labeled with cy2, visualized in green), and Giantin (labeled with cy5, blue channel but shown in red) were labelled with primary antibodies followed by cy2- and cy5-conjugated secondary antibodies, respectively. For easier visualization of the overlay channel, the pictures were elaborated after confocal microscopy acquisition, to split channels by rendering red the cy5 (blue) channel.

For mitochondrial staining, cells were incubated with Mitotracker Red CMXRos (500 nM) 20 min at 37 °C, before fixing with PFA and cold methanol 5 min on ice. Quantitative analysis was performed on a minimum of 30 cells by setting the same threshold of fluorescence intensity in all the samples analyzed. The mitochondrial network was then described in numbers of objects (ImageJ software) and object volume using the 3D object counter available in the software Fiji (http://www.fiji.sc).

For Tfr-Alexa 594 staining, the cells were incubated 30 min at 37 °C in complete medium before proceeding with immunofluorescence. Nuclei were stained by using DAPI (1:1000) in PBS.

Pearson’s Correlation Coefficient (PCC) was employed to quantify colocalization [60] between APP and Giantin (as well as Tfr), and was determined in at least 25 cells from three different experiments. PCC was calculated in regions of APP and reference protein co-presence as previously described [10].

The PCC was then calculated in the defined regions for the images of interest. Immunofluorescences were analyzed by the confocal microscope LSM 700 Zeiss equipped with an oil immersion 63 × 1.4 NA Plan Apochromat objective (or 40×), and a pinhole size of one airy unit.

### 4.5. Direct ELISA for Detection of Aβ in Conditioned Medium from Human Skin Fibroblasts

High binding plates with 96 flat-bottomed wells (Corning, Amsterdam, The Netherlands) were coated with conditioned medium from healthy and AD human skin fibroblasts, or BSA as a negative control, and incubated at 4 °C overnight. After washing in PBS, residual binding sites were blocked for 1 h at 37 °C with 200 μL of blocking buffer (2% FCS, 1 mg/mL BSA, in PBS). Wells were incubated with 0.002 μg/mL of anti-β-Amyloid monoclonal antibodies (6E10 or 4G8) or non-immune control immunoglobulins for 1 h at room temperature. Each well was washed three times with wash buffer (0.5% Tween in PBS). HRP-conjugated goat anti-mouse secondary antibody (1:5000) was added for 1 h at room temperature. After washing in PBS, OPD (o-phenylenediamine dihydrochloride) substrate solution was added and absorbance was detected at 490 nm on an ELISA plate reader (Bio-Rad). Binding affinity was determined by subtracting background absorbance (BSA wells).

For experiments with NSC47924, healthy and fAD fibroblasts were incubated for 24 h at 37 °C. After treatment the conditioned media were collected and stored at −20 °C.

### 4.6. Statistical Analysis

Data are expressed as mean ± SD. All statistical analyses using two-way ANOVA and histograms were completed with excel software. Differences were considered statistically significant when *p* < 0.05.

## Figures and Tables

**Figure 1 jpm-10-00232-f001:**
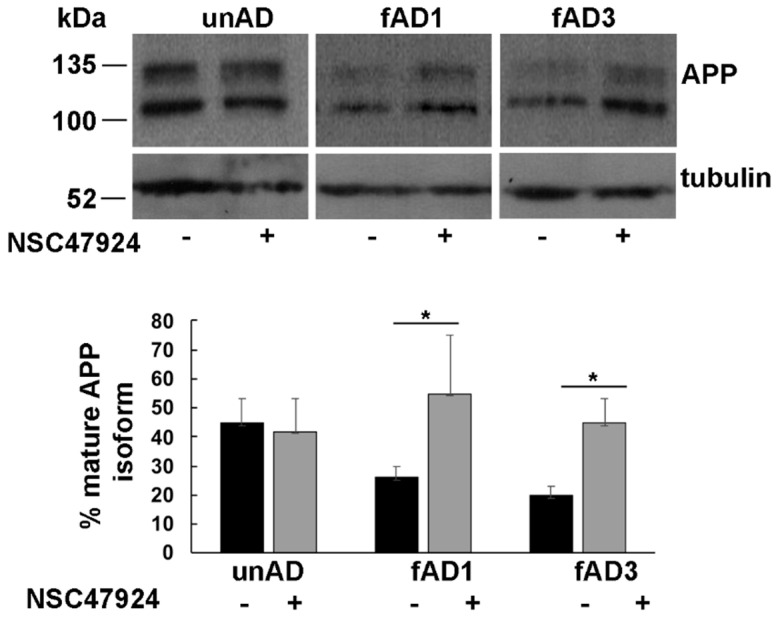
Inhibition of 37/67kDa LR by NSC47924 restored the amyloid precursor protein (APP) isoform ratio in fAD fibroblasts. Cultured fAD1, fAD3 and unAD fibroblasts were scraped in lysis buffer and 40 μg of total proteins were subjected to SDS-PAGE. APP was revealed by Western blotting on PVDF (Polyvinylidene Difluoride membranes) and hybridization with the A8717 antibody. fAD1, fAD3, and unAD fibroblasts were treated with NSC47924 for 24 h. Protein levels of mature APP were calculated by densitometric analysis with ImageJ software and expressed as a percentage. The plot shows the percentage of mature APP isoform, using 100% as the sum values of immature plus mature isoform with respect to tubulin as a loading control, under treated or not treated conditions. Data are expressed as the means + SD of three independent experiments (* *p* < 0.05).

**Figure 2 jpm-10-00232-f002:**
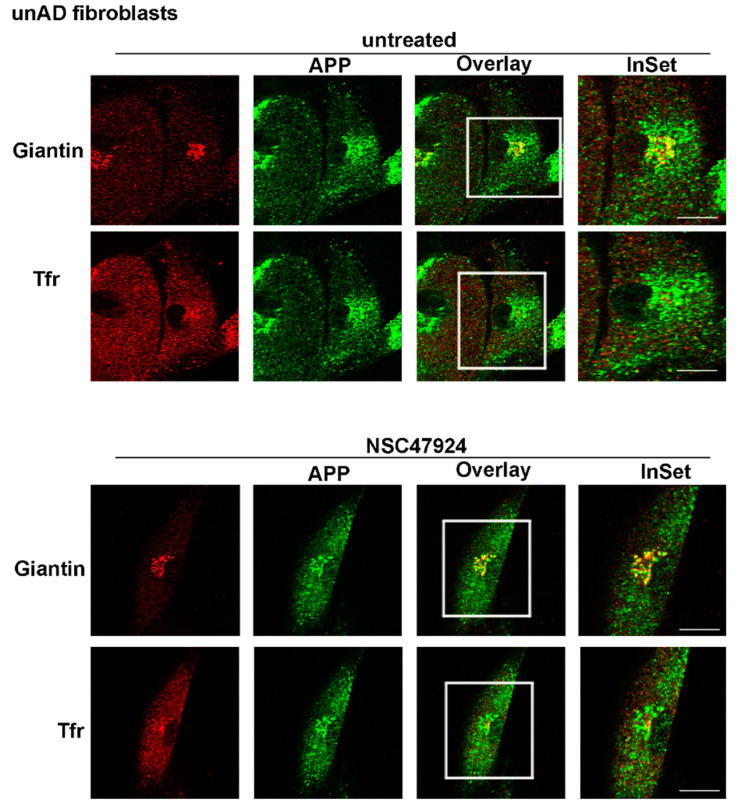
APP is localized in the Golgi apparatus in control unAD fibroblasts. Cells were grown on coverslips, fixed in PFA 4% and permeabilized in 0.1% TX-100 for 30 min, then they were stained by double immunolabeling with A8717 rabbit antibody (1:500) and Giantin (1:50), followed by cy2- and cy5-conjugated secondary antibodies, to label APP (green) and Golgi (red), respectively. Tfr Alexa-594 (red) in the cell culture media was used to label recycling endosomes. Colocalization between APP and the different markers was then measured as indicated in the methods section. The color channels in the triple image acquisition, have been split into green, red and blue. The blue channel is attributed to the cy5-conjugated secondary antibody, for easier visualization of the overlay, it has been rendered red after channel splitting and picture acquisition. Images from NSC47924 treatment are shown in the bottom panel. Scale bars, 10 μm.

**Figure 3 jpm-10-00232-f003:**
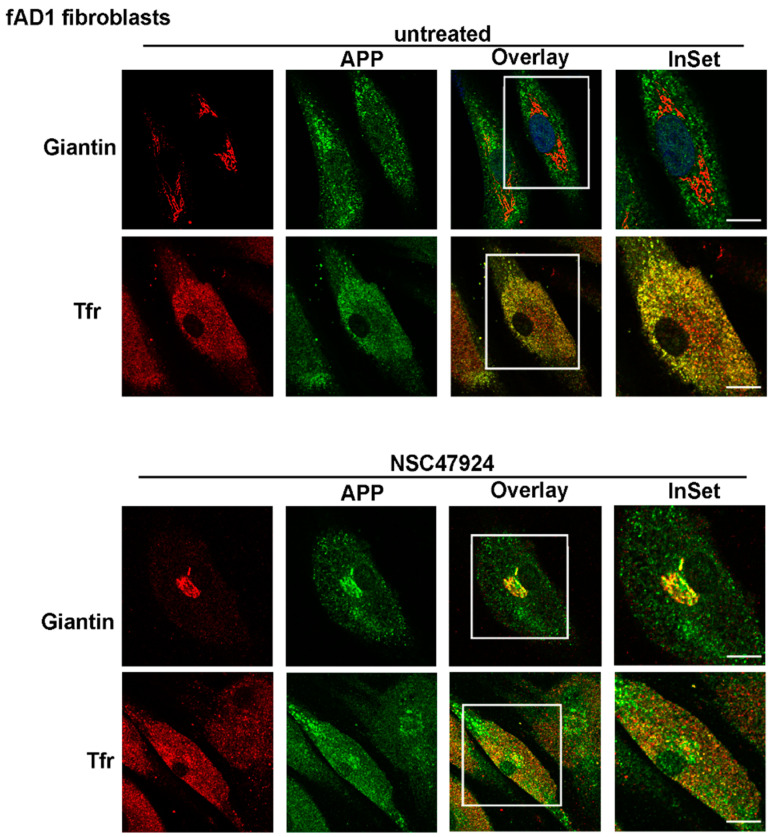
APP is mainly localized in Tfr-positive recycling endosomes in fAD1 fibroblasts. FAD1 cells were grown and processed for immunofluorescence as in Figure 2. Note the absence of APP in the Golgi apparatus and the presence in recycling endosomes in untreated fAD fibroblasts. Intracellular APP localization is reversed by NSC47924 incubation. Scale bars, 10 μm.

**Figure 4 jpm-10-00232-f004:**
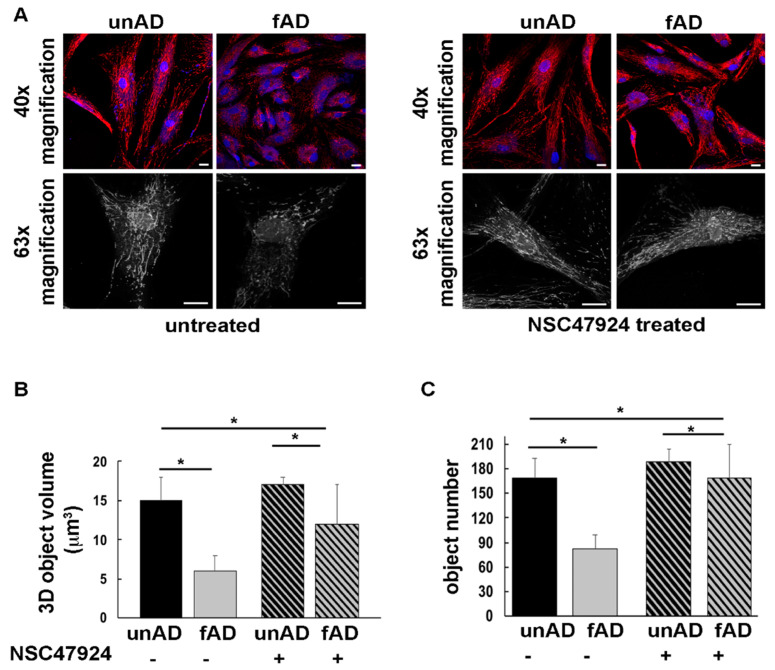
The mitochondrial network is altered in fAD fibroblasts and is restored by 37/67kDa LR inhibition by NSC47924. (**A**) UnAD (control) and fAD cells were seeded on glass coverslips and stained with the Mitotracker Red CMXRos probe for 20 min at 37 °C, then they were fixed and processed for immunofluorescence. Serial confocal sections were collected from top to bottom of one selected fAD cell line. Representative images of a single confocal plane (40× magnification, upper panels) and 3D reconstructions (black and white images, bottom panel) of Z-stack acquired to higher magnification (63×) are shown. Scale bars, 10 μm. (**B**) The average mitochondrial volume (expressed as 3D object volume, μm^3^) is significantly higher in unAD cells compared with fAD cells. The bars show mean values + SD of two fAD and two unAD cell cultures. Fifteen randomly selected cells for each experimental condition were analyzed. NSC47924 was used as indicated in the methods section. (**C**) The number of mitochondria is significantly higher in unAD fibroblasts compared to fAD cells. Inhibition of the receptor by NSC47924 restored the parameters to that of unAD control ones (* *p* < 0.05).

**Figure 5 jpm-10-00232-f005:**
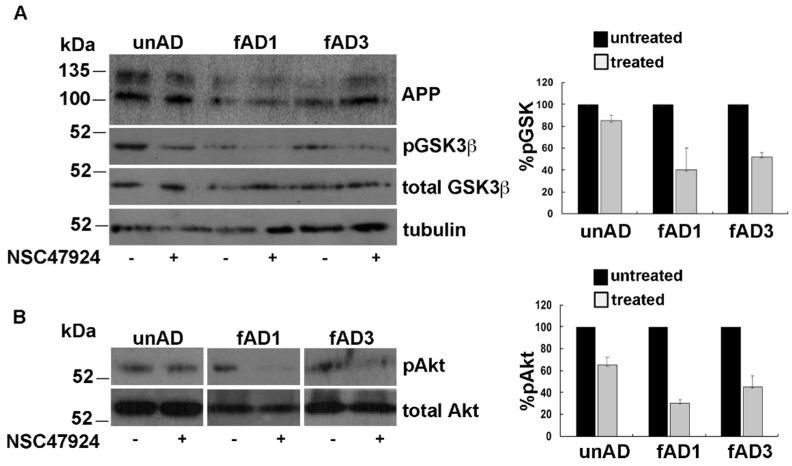
NSC47924 induces inactivation of Akt with the consequent decrement of pGSK3β. (**A**) Total cell lysates (40 μg) from unAD and fAD cells treated or not with NSC47924, were loaded on gels. Both total and pGSK3β levels were revealed by using anti-GSK3β and anti-phospho-Ser9 GSK3β antibody, respectively. (**B**) The membranes were treated as in (**A**) with the exception that here they were hybridized with anti-Akt and pAkt antibodies. The plots on the right show the percentage of pGSK and pAkt, using as 100% the values of pGSK3β and pAkt in the absence of the inhibitor. Data are expressed as the means + SD of three independent experiments (all data were statistically significant).

**Figure 6 jpm-10-00232-f006:**
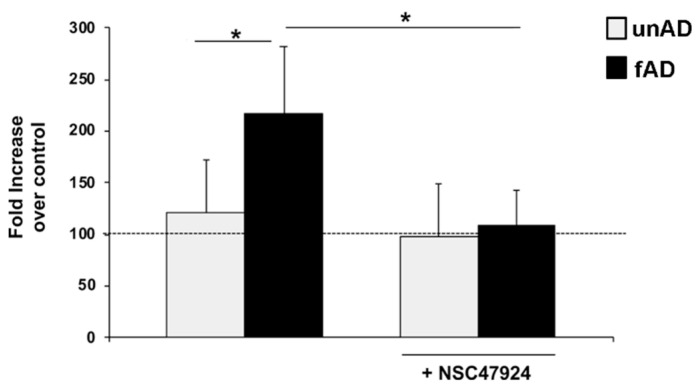
Aβ secreted levels are higher in fAD cell culture media and are decreased by inhibitor treatment. Conditioned medium from healthy (gray) and fAD (black) fibroblasts was incubated with an anti-Aβ monoclonal antibody (6E10) or non-immune control immunoglobulins IgG. The bound anti-Aβ antibody was revealed by OPD staining; the absorbance at 490 nm was measured. Anti-Aβ antibody binding to BSA-coated wells was subtracted to obtain a specific binding. Results are expressed as a percentage increase of anti-Aβ antibody absorbance value over non-immune control IgG, using as 100% the expression value in control conditions. Values represent the mean ± SEM of three experiments carried out in triplicate; (* *p* < 0.05).

**Table 1 jpm-10-00232-t001:** List of DNA variants identified in *APP*, *PSEN1* and *PSEN2* by Sanger sequencing.

Gene	Exon	c.DNA	Protein	Genotype	rs	Clinvar	Varsome (DANN Score)	ACMG	Allelic Frequency *	HSF
*APP*	IVS10	c.1402–106A > G	-	HOM	rs440666	NA	0.29	Benign	0.71	-
	IVS11 A11	c.1530+79C > G	-	HOM	rs2251337	NA	0.39	Benign	0.88	-
*PSEN1*	IVS6	c.480+223A > G	-	HOM	rs214269	NA	0.58	Benign	0.73	No significant splicing motif alteration detected This mutation has probably no impact on splicing
	UTR	c.*2314delT	-	HOM	rs895744391	NA	-	Benign	0.01	-
*PSEN2*	4	c.69T > C	p.Ala23=	HOM	rs11405	Benign	0.39	Benign	0.76	-
	4	c.129C > T	p.Asn43=	HOM	rs6759	Benign	0.48	Benign	0.48	-
	IVS4	c.142–42G > A	-	HOM	rs1295643	NA	0.74	Benign	0.49	No significant splicing motif alteration detected This mutation has probably no impact on splicing
	IVS4	c.142–29T > C	-	HOM	rs1295644	NA	0.35	Benign	0.71	Alteration of an intronic ESS site Probably no impact on splicing
	5	c.261C > T	p.His87=	HOM	rs1046240	Benign	0.65	Benign	0.50	-
	IVS7	c.498+30G > C	-	HOM	rs2236910	NA	0.59	Benign	0.77	Creation of an intronic ESE site Probably no impact on splicing
	IVS9	c.887–24T > C	-	HOM	rs2802267	NA	0.41	Benign	0.75	Alteration of an intronic ESS site. Probably no impact on splicing. Creation of an intronic ESE site Probably no impact on splicing
	IVS11	c.1073–86A > G	-	HOM	rs10753428	NA	0.37	Benign	0.75	Creation of an intronic ESE site Probably no impact on splicing
	IVS12	c.1191+24G > A	-	HOM	rs2855562	NA	0.53	Benign	0.54	No significant splicing motif alteration detected This mutation has probably no impact on splicing

rs: reference SNP; DANN: Deleterious Annotation of Genetic Variants; ACMG: American College of Medical Genetics; HSF: Human Splicing Finder (http://www.umd.be/HSF/HSF.shtml); IVS: InterVening Sequence; HOM: homozygous; NA: not available; UCV: unclassified variant. * based on gnomAD Genomes, version: 2.1.1.

**Table 2 jpm-10-00232-t002:** Mature/Immature APP isoform ratio.

	Untreated	NSC47924 (20 μM)
**unAD**	0.42 ± 0.08	0.45 ± 0.1
**fAD1**	0.26 ± 0.04	0.55 ± 0.2
**fAD3**	0.20 ± 0.03	0.45 ± 0.08

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
