# Peer review of "Inhibition of 37/67kDa Laminin-1 Receptor Restores APP Maturation and Reduces Amyloid-β in Human Skin Fibroblasts from Familial Alzheimer’s Disease"

_jpm, 2020, doi:10.3390/jpm10040232_

Round 1

Reviewer 1 Report

In this study Bhattacharya et al describe that inhibition of 37/67 kD laminin-1 receptor by NSC47924 can improve Alzheimer’s related cellular phenotypes in two familial fibroblast models of the disease.  The authors analyze APP maturation and trafficking as well as mitochondrial phenotypes, Akt signaling and GSK3b signaling, and A-beta secretion.

Overall, the results are consistent with an improvement in Alzheimer’s phenotypes; however, some improvements in statistical analysis and additional controls are necessary.

  • The statistical test (Student’s t-test) for significance used throughout the study is inappropriate. In order to assess changes between 2 or more genotypes +/- treatment the appropriate statistical test would be a 2-way ANOVA.
  • The labeling of the y-axis figure 4B is unclear. Please indicate appropriate volume units like square microns or square pixels.
  • In Figure 5 a quantification and statistical analysis of differences in p-Akt and p-GSK3b levels is missing.
    1. Please show a tubulin loading control
    2. Please show Akt WBs from FAD1
  • In the discussion it is not clear:
    1. What is the hypothesis behind testing a compound that induces APP mislocalization and inhibits its maturation (Ref 10) to rescue it under FAD conditions, and also why you think you were able to achieve this rescue. This appears to be the opposite effect of what the authors observed before
    2. The authors have observed an increase of Akt phosphorylation previously under compound treatment (Ref 10). It requires more discussion why the opposite effect is observed in this study

Once these points are addressed I can recommend this study for publication

Author Response

Point-by-point reply to reviewers

 Rev#1

Comments and Suggestions for Authors

In this study Bhattacharya et al describe that inhibition of 37/67 kD laminin-1 receptor by NSC47924 can improve Alzheimer’s related cellular phenotypes in two familial fibroblast models of the disease.  The authors analyze APP maturation and trafficking as well as mitochondrial phenotypes, Akt signaling and GSK3b signaling, and A-beta secretion.

Overall, the results are consistent with an improvement in Alzheimer’s phenotypes; however, some improvements in statistical analysis and additional controls are necessary.

  1. The statistical test (Student’s t-test) for significance used throughout the study is inappropriate. In order to assess changes between 2 or more genotypes +/- treatment the appropriate statistical test would be a 2-way ANOVA.

Response: we thank the referee for the suggestion to use 2-way ANOVA for assessing statistical changes after treatment of cells with inhibitor. Accordingly, we have now replaced the statistics throughout the manuscript and methods section (pag 20, line 464).

  1. The labeling of the y-axis figure 4B is unclear. Please indicate appropriate volume units like square microns or square pixels.

Response: The y-axis of Figure 4B has been now labeled by reporting the 3D object volume expressed as square microns. Accordingly, the Figure 4B, and relative legend, have been replaced (pag 9).

  1. In Figure 5 a quantification and statistical analysis of differences in p-Akt and p-GSK3b levels is missing.
  1. Please show a tubulin loading control

Response: Figure 5 (pag 11) has been now replaced. Quantification and statistical analysis has been now performed on p-Akt and p-GSK3b gels together with the inclusion of the tubulin wb as loading control for procedure.

  1. Please show Akt WBs from FAD1

Response: Figure 5B, Akt wbs from fAD1 has been now shown for completeness of results.

  1. In the discussion it is not clear:
    1. What is the hypothesis behind testing a compound that induces APP mislocalization and inhibits its maturation (Ref 10) to rescue it under FAD conditions, and also why you think you were able to achieve this rescue. This appears to be the opposite effect of what the authors observed before

Response: we thank the referee for this observation that give us the opportunity to better explain the rationale behind our experimental approach (see pag 13, line 303-311 of discussion in the revised text). We have the possibility to inhibit 37/67kDa LR with different compounds that differ for at least one element in the structure (Pesapane et al., 2015). In two previous already published studies, we have tested two analogs: NSC47924 (Ref21 of previous version of ms, now Ref 22) and NSC48478 (Ref10), which differ by a O-CH3 or Cl on the naphthol ring, respectively. The first molecule, NSC47924 has been able to control Prion Protein trafficking in mouse neuronal cell line. The second one, NSC48478, has been shown to control trafficking of APP in mouse neuronal cells with consequences on its maturation and intracellular signaling. Protein maturation through the secretory pathway is strictly related to its intracellular trafficking. Since APP maturation is crucial for correct protein processing and coordination between the non-amyloidogenic and amyloidogenic pathway, we decided to test the effects of the two inhibitors in pathological conditions, human cells derived from AD affected individuals. After pilot experiments on APP using both analogs in human cells, only NSC47924 has been found to exert effects on trafficking and maturation of APP, restoring the “correct” ones that were lost in affected cells. Therefore, we decided to report only the active molecule NSC47924 in the present study. Thus, our data are not in contrast with ones we obtained with NSC48478 in a different cell line (neuronal mouse), under different conditions.

    1. The authors have observed an increase of Akt phosphorylation previously under compound treatment (Ref 10). It requires more discussion why the opposite effect is observed in this study

Response: the reply to this referee’s observation in some way recalls the response above. The Akt phosphorylation previously tested in Ref10 (Bhattacharya et al., 2020), refers to experiments done in mouse neuronal cell line and with a different analog inhibitor of LR. Therefore, the effects observed in human fibroblasts and reported in the present manuscripts, likely are not in contrast with previous ones.

Once these points are addressed I can recommend this study for publication

Reviewer 2 Report

The paper from Bhattacharya et al reports the anti-amyloidogenic effect of laminin 1 receptor in human fibroblasts with fAD mutations. LR inhibition by the specific inhibitor reduces Abeta generation and consequent Abeta release by redirecting APP trafficking from the endosomal recycling pathway to the Golgi network. Also, the LR inhibitor improves mitochondrial defects typical of fAD fibroblasts.

Main points

- It is well known that APP trafficking is unique in neurons, most probably not even comparable to neuronal cell lines. Since very few reports from the literature are focused on neuronal APP trafficking, of interest for AD, all the relevant literature should be better included in the Discussion.

-Why fibroblasts instead of iPS-derived neurons were used in this study? The potential advantages of the model of choice should be briefly discussed.

-INTRO: the authors state that APP is the most common cause of AD. However, the amyloid hypothesis has been recently questioned by the failure of most of the APP-centred clinical trials.

- Abeta can be produced intracellularly in lysosomes, plus not all the Abeta generated is released in culture medium. Thus, intracellular Abeta from fibroblast extracts should be measured, as a control, as well as a counterproof of the prevalent APP shuttling to the Golgi upon LR inhibitor incubation.

- Reduced level of mature APP could mainly (or also) reflect reduced posttranslational modification (glycosylation) typical of AD, instead of altered Golgi trafficking. Please, discuss.

- Why the authors studied the effect of LR inhibitor on mitochondria? Where is the link with Abeta or APP trafficking to the Golgi?

- Fig. 2: Why Tfr and giantin are both reported red as colour? APP pictures (green channel) are identical: was it a triple immunostaining (APP/TfR/giantin)? Please, explain, and add details to the legend and Methods section.

-Fig.3: fAD fibroblast + LR inhibitor, giantin staining: cell morphology is reminiscent of a damaged /degenerating cell, not fitting the hypothesized recovery following LR inhibitor incubation.

-Fig.4: fAD+LR inhibitor (40x): DAPI positive nuclei are not clearly visible in this picture. Please, replace.

Author Response

Rev#2

Comments and Suggestions for Authors

The paper from Bhattacharya et al reports the anti-amyloidogenic effect of laminin 1 receptor in human fibroblasts with fAD mutations. LR inhibition by the specific inhibitor reduces Abeta generation and consequent Abeta release by redirecting APP trafficking from the endosomal recycling pathway to the Golgi network. Also, the LR inhibitor improves mitochondrial defects typical of fAD fibroblasts.

Main points

- It is well known that APP trafficking is unique in neurons, most probably not even comparable to neuronal cell lines. Since very few reports from the literature are focused on neuronal APP trafficking, of interest for AD, all the relevant literature should be better included in the Discussion.

Response: we agree with the referee’s comment. Including the references concerning APP trafficking in neurons, we hope the manuscript has been improved. The following references have been added:

- Jing Zhi A Tanet al. Distinct anterograde trafficking pathways of BACE1 and amyloid precursor protein from the TGN and the regulation of amyloid-β production - Molecular Biology of Cell – 2019 https://doi.org/10.1091/mbc.e19-09-0487

- Das U. et al. –Activity-induced convergence of APP and BACE-1 in acidic microdomains via an endocytosis-dependent pathway – Neuron – 2013,  https://doi.org/10.1016/j.neuron.2013.05.035

- Wang X et al., Modifications and trafficking of APP in the pathogenesis of Alzheimer’s disease,” Frontiers in Molecular Neuroscience, vol. 10. Frontiers Media S.A., Sep. 15, 2017, doi: 10.3389/fnmol.2017.00294

- DelBove CE, Deng XZ, Zhang Q. The Fate of Nascent APP in Hippocampal Neurons: A Live Cell Imaging Study. ACS Chem Neurosci. 2018 Sep 19;9(9):2225-2232. doi: 10.1021/acschemneuro.8b00226. Epub 2018 Jun 21.

Accordingly, we also revised the discussion section considering these new references (pag 14, line 325-348).

-Why fibroblasts instead of iPS-derived neurons were used in this study? The potential advantages of the model of choice should be briefly discussed.

Response: We fully agree with the referee (see description at pag 13, lines 284-293). Fibroblasts have been employed as in vitro model for neurological diseases (G. P. Connolly, “Fibroblast models of neurological disorders: fluorescence measurement studies,” Trends in Pharmacological Sciences, vol. 19, no. 5, pp. 171–177, 1998), and particularly for AD (L. Gasparini, M. Racchi, G. Binetti et al., “Peripheral markers in testing pathophysiological hypotheses and diagnosing Alzheimer's disease,” The FASEB Journal, vol. 12, no. 1, pp. 17–34, 1998); (S. Govoni, L. Gasparini, M. Racchi, and M. Trabucchi, “Peripheral cells as an investigational tool for Alzheimer's disease,” Life Science, vol. 59, no. 5-6, pp. 461–468, 1996); (Altered proteolysis in fibroblasts of Alzheimer patients with predictive implications for subjects at risk of disease. Mocali A, Della Malva N, Abete C, Mitidieri Costanza VA, Bavazzano A, Boddi V, Sanchez L, Dessì S, Pani A, Paoletti F. Int J Alzheimers Dis. 2014;2014:520152. doi: 10.1155/2014/520152. Epub 2014 May 18). One major advantage of using fibroblasts in drug discovery studies is the high availability of fibroblasts that can be isolated from skin biopsies; furthermore, their cultivation, propagation and cryoconservation are uncomplicated with respect to nutritional requirements and viability in culture. Specifically, fibroblasts from Alzheimer’s disease affected individuals are the most convenient and easily available cell model closest to human patients. This model could help us to elucidate the mechanism of the small compound on APP trafficking and processing, before developing a more complex system, like iPSC lines and consequently iPSC-derived neurons. Since fibroblasts are the most commonly used primary somatic cell type for the generation of induced pluripotent stem cells, we hope to obtain in short time a better characterization of biological processes related to NSC47924 action on FAD cell patient, to planning subsequently, a reprogramming and differentiation of patient fibroblasts. Comparison of the effects of different LR inhibitor molecules on different cell types that may be considered to be non-relevant for the disease, such as fibroblasts, or more relevant to the disease, such as neurons differentiated from iPSCs, will allow us to develop more predictive in vitro systems for drug discovery. Our results obtained in fibroblasts represent the first step to reinforce the value of utilizing human iPSCs in drug discovery to improve translational predictability.

-INTRO: the authors state that APP is the most common cause of AD. However, the amyloid hypothesis has been recently questioned by the failure of most of the APP-centred clinical trials.

- Abeta can be produced intracellularly in lysosomes, plus not all the Abeta generated is released in culture medium. Thus, intracellular Abeta from fibroblast extracts should be measured, as a control, as well as a counterproof of the prevalent APP shuttling to the Golgi upon LR inhibitor incubation.

Response: we strongly agree with the referee’s comment, so much so that we had previously performed this analysis. We did not put it in the previous version of the manuscript for a problem of length. Therefore, in the revised Figure S1 we now show the IF analysis of Ab in fAD cells. Specifically, by using 6E10 antibody, directed to the 1-16 epitope of the amyloid, we performed a double IF analysis with anti-Giantin antibody, marker of the Golgi.

As shown in the right panels of Figure S1, we can see that, beside Golgi relocation of APP after NSC47924 treatment (compare versus untreated), any accumulation of Ab was detectable in the cells after treatment with LR inhibitor. This data suggests that the reduction of amyloid beta in the cell culture media, was likely due to impairment of Ab production rather than to a lack of its secretion (see revised text, results section pag 12, line 272-274; discussion section pag 16, lines 364-367).

- Reduced level of mature APP could mainly (or also) reflect reduced posttranslational modification (glycosylation) typical of AD, instead of altered Golgi trafficking. Please, discuss.

Response: (see pag 14 lines 325-348) It has been observed that APP modifications and trafficking are mutually regulated, contributing in turn to the modulation of Aβ generation (X. Wang, X. Zhou, G. Li, Y. Zhang, Y. Wu, and W. Song, “Modifications and trafficking of APP in the pathogenesis of Alzheimer’s disease,” Frontiers in Molecular Neuroscience, vol. 10. Frontiers Media S.A., Sep. 15, 2017, doi: 10.3389/fnmol.2017.00294). The acquisition of APP post-translational modifications mainly occurs during the passage through the secretory pathway (J. Walter, and C. Haass, “Posttraslational modifications of Amyloid Precursor Protein,” Methods Molecular Medicine, 2000, vol. 32, pp. 149–168, doi: 10.1385/1-59259-195-7:149), where the protein can undergo glycosylation which starts in the ER to proceed through the Golgi, where sulfation and phosphorylation occur. Since reduced level of mature APP in AD-affected cells was accompanied by a mislocalization of APP from the Golgi and consequent accumulation into recycling endosomes (Figure 3, Figure S1 and new S2), we speculated that APP was not able to be correctly modified for maturation probably because of its inability to transit through the secretory pathway and to reach the Golgi apparatus, where the majority of APP modifications occurs.

Therefore, the possibility to control APP trafficking (by inhibiting the receptor LR, as well as we have previously done in neuronal cells, Ref10 of previous version, Bhattacharya et al., 2020) would be reflected in APP maturation changing. Our hypothesis has been confirmed by showing that after inhibitor treatment, APP recovered its Golgi localization and its molecular weight at the same level of unaffected cells (Figure 2 and 3).

- Why the authors studied the effect of LR inhibitor on mitochondria? Where is the link with Abeta or APP trafficking to the Golgi?

Response: (pag 15, lines 338-348) Processing of APP by a-, b- and g-secretases generating APP-CTFs (C-terminal fragments) and Ab, is strictly related to the subcellular localization and trafficking of APP. Because the secretases have different distribution within the cells, the itinerary followed by APP is crucial for determining the amount of APP fragments. Furthermore, CTFs have been found localized to subcellular microdomains between the ER and mitochondria (Del Prete D, Suski JM, Oules B, Debayle D, Gay AS, Lacas- Gervais S et al (2017) Localization and processing of the amyloid- beta protein precursor in mitochondria-associated membranes. J Alzheimers Dis 55:1549–1570), and they can lead, as well as Ab, to mitochondrial dysfunction and morphology alteration in in vivo and in vitro AD study models (Anandatheerthavarada HK, Devi L Amyloid precursor protein and mitochondrial dysfunction in Alzheimer’s disease. Neuroscientist 2007 13:626–638; Reddy PH, Beal MF Amyloid beta, mitochondrial dysfunction and synaptic damage: implications for cognitive decline in aging and Alzheimer’s disease. Trends Mol Med 2008 14:45–53; Vaillant-Beuchot L, et al. Accumulation of amyloid precursor protein C-terminal fragments triggers mitochondrial structure, function, and mitophagy defects in Alzheimer’s disease models and human brains. Acta Neuropathologica 2020). According to findings in literature, we observed altered mitochondrial network in fAD cells. Therefore, we speculated that if in AD-affected cells we would be able (by using LR inhibitor) to reverse the subcellular localization of APP and, consequently its trafficking to that of healthy cells, we should have observed a restoring of cellular phenotype resembling the one of healthy conditions, including mitochondrial network phenotype.

- Fig. 2: Why Tfr and giantin are both reported red as colour? APP pictures (green channel) are identical: was it a triple immunostaining (APP/TfR/giantin)? Please, explain, and add details to the legend and Methods section.

Response: We apologize for the concise description we made for Figure 2 in the previous text version. We realize that, in this form, it is confusing to the readers. APP-cy2/Tfr-594/Giantin-cy5 were labeled by a triple immunostaining using different secondary antibodies. The colour channels have been splitted in green, red and blue. The blue channel attributed to the third fluorophore (cy5), for an easier visualization of the overlay, has been rendered red after splitting. Figure 2 legend (pag 7) and methods section (pag 18, line 428-434) have been replaced accordingly.

-Fig.3: fAD fibroblast + LR inhibitor, giantin staining: cell morphology is reminiscent of a damaged /degenerating cell, not fitting the hypothesized recovery following LR inhibitor incubation.

Response: we agree with the referee. The picture reporting Giantin staining gives the feeling of showing a damaged cell. This is not the case in the majority of our experiments. We have now replaced Giantin staining in the Figure 3 (pag 8).

-Fig.4: fAD+LR inhibitor (40x): DAPI positive nuclei are not clearly visible in this picture. Please, replace.

Response: done

Reviewer 3 Report

This manuscript by Antaripa Bhattacharya and colleagues reports on a specific inhibitor of laminin-1 receptor restores the localization and maturation of APP through AKT-GSK3b pathway. In brief, the ratio of mature and immature form of APP was reduced in the fibroblast cell line from AD compared to healthy control. In addition, APP was mainly localized in the Golgi apparatus in the AD fibroblast while APP is normally distributed in the endosome. These abnormalities in the AD fibroblasts was corrected by the administration of laminin-1 receptor inhibitor. The idea that APP localization and maturation are deregulated in the cells from patient with AD and an inhibitor of laminin-1 receptor could be a therapeutic for AD through restoring APP maturation is potentially interesting, but the manuscript lacks some information and needs some improvements.

  1. The authors performed all the experiments using fibroblast cell lines obtained from AD patients or healthy control. However, the main elements in the AD pathogenesis is the loss of “neuronal cells” in the brain but not the fibroblasts. The authors should justify why they use fibroblasts but not the neuronal cells, otherwise, they should use neuronal cells, e.g. iPS cells, from the patients or healthy control.
  2. The authors somehow exclude the data of experiments using fAD2 cell line. They should describe the reason for excluding the fAD2 data from this manuscript.
  3. The authors analyzed fibroblast cell lines from familial AD patients. However, the results show that no pathogenic variants were found in the APP, PSEN1 and PSEN2 genes of AD cell line. I was wondering how this patient (fAD1) was diagnosed as “familial” AD without genetic mutation. In addition, the information about age, sex and AD stage lacks in this manuscript. These information would be needed to be described in the main text.
  4. The authors described as “we found that the mature/immature APP ratio was significantly lower in both fAD fibroblast cell lines” in page 5 line 110-111. This result (with statistical analysis?) should be reflected in the Figure 1.
  5. The authors used Transferrin-conjugated Alexa 594 and secondary antibodies conjugated with Alexa-488 and Alexa-546 for immunostaining according to the description in the method section. I have doubts about the possible crosstalk artifacts since there is overlap in the spectra between Alexa-488 and Alexa-546 or between Alexa-546 and Alexa-594. In the Figure2, co-immunostaining of these three dyes was seemed to be performed since cell shape looks same. Have you checked your microscope setup carefully to be sure that there is no possibility for crosstalk? Otherwise, I recommend you to perform same immunocytochemical experiment (Figure3 as well as Figure2) using another combination of fluorescence-conjugated antibodies.
  6. The authors described that all statistical analyses were performed using Student’s t-test. However, Student’s t-test cannot be used for a comparison of more than two groups. The authors should perform statistical analysis using the test other than Student’s t-test.

Author Response

Rev#3

Comments and Suggestions for Authors

This manuscript by Antaripa Bhattacharya and colleagues reports on a specific inhibitor of laminin-1 receptor restores the localization and maturation of APP through AKT-GSK3b pathway. In brief, the ratio of mature and immature form of APP was reduced in the fibroblast cell line from AD compared to healthy control. In addition, APP was mainly localized in the Golgi apparatus in the AD fibroblast while APP is normally distributed in the endosome. These abnormalities in the AD fibroblasts was corrected by the administration of laminin-1 receptor inhibitor. The idea that APP localization and maturation are deregulated in the cells from patient with AD and an inhibitor of laminin-1 receptor could be a therapeutic for AD through restoring APP maturation is potentially interesting, but the manuscript lacks some information and needs some improvements.

  1. The authors performed all the experiments using fibroblast cell lines obtained from AD patients or healthy control. However, the main elements in the AD pathogenesis is the loss of “neuronal cells” in the brain but not the fibroblasts. The authors should justify why they use fibroblasts but not the neuronal cells, otherwise, they should use neuronal cells, e.g. iPS cells, from the patients or healthy control.

Response: We fully agree with the referee’s observation (see discussion at pag 13, lines 284-293). Fibroblasts have been employed as an in vitro model for neurological diseases (G. P. Connolly, “Fibroblast models of neurological disorders: fluorescence measurement studies,” Trends in Pharmacological Sciences, vol. 19, no. 5, pp. 171–177, 1998; Auburger G., Mol Neurobiol, 46:20-27, 2012), and particularly for AD (L. Gasparini, M. Racchi, G. Binetti et al., “Peripheral markers in testing pathophysiological hypotheses and diagnosing Alzheimer's disease,” The FASEB Journal, vol. 12, no. 1, pp. 17–34, 1998); (S. Govoni, L. Gasparini, M. Racchi, and M. Trabucchi, “Peripheral cells as an investigational tool for Alzheimer's disease,” Life Science, vol. 59, no. 5-6, pp. 461–468, 1996); (Altered proteolysis in fibroblasts of Alzheimer patients with predictive implications for subjects at risk of disease. Mocali A, Della Malva N, Abete C, Mitidieri Costanza VA, Bavazzano A, Boddi V, Sanchez L, Dessì S, Pani A, Paoletti F. Int J Alzheimers Dis. 2014;2014:520152. doi: 10.1155/2014/520152. Epub 2014 May 18)

One major advantage of using fibroblasts in drug discovery studies is the high availability of fibroblasts that can be isolated from skin biopsies; furthermore, their cultivation, propagation and cryoconservation are uncomplicated with respect to nutritional requirements and viability in culture. Specifically, fibroblasts from Alzheimer’s disease affected individuals are the most convenient and easily available cell model closest to human patients.

This model could help us to elucidate the mechanism of the small compounds on APP trafficking and processing, before developing a more complex system, like iPSC lines and consequently iPSC-derived neurons. Since fibroblasts are the most commonly used primary somatic cell type for the generation of induced pluripotent stem cells, we hope to obtain in short time a better characterization of biological processes related to NSC47924 action on FAD cell patient, so to planning a reprogramming and differentiation of patient fibroblasts. Comparison of the effects of different LR inhibitor molecules on different cell types that may be considered to be non-relevant for the disease, such as fibroblasts, or more relevant to the disease, such as neurons differentiated from iPSCs, will allow us to develop more predictive in vitro systems for drug discovery. Our results obtained in fibroblasts represent the first step to reinforce the value of utilizing human iPSCs in drug discovery to improve translational predictability.

  1. The authors somehow exclude the data of experiments using fAD2 cell line. They should describe the reason for excluding the fAD2 data from this manuscript.

Response: (see new Figure S2 in supplementary material). We agree with the referee that excluding fAD2 data could not be rationale. Our explanation is that the entire set of experiments give the same results for fAD1, fAD2 and fAD3. Since fAD2 cell line carries the same genetic mutation (PSEN2 M239V) as fAD3, in order to make the manuscript leaner, we thought to not show fAD2. However, because we have performed the main experiments on APP trafficking and maturation in fAD2, we have added a new Figure S2 showing fAD2 IF and Wblots in the revised version.

  1. The authors analyzed fibroblast cell lines from familial AD patients. However, the results show that no pathogenic variants were found in the APP, PSEN1 and PSEN2 genes of AD cell line. I was wondering how this patient (fAD1) was diagnosed as “familial” AD without genetic mutation. In addition, the information about age, sex and AD stage lacks in this manuscript. These information would be needed to be described in the main text.

Response: pages 17-18, line 404-414. Fibroblasts were derived directly from the skin punch biopsy of the Italian patients carrying (fAD2 and fAD3, male with onset AD 48 year, and female with onset AD 45 year, respectively) or not (fAD1, female, with unknown onset AD) the PSEN2 M239V mutation. All fibroblasts were derived from patients with a clinical diagnosis of probable familial early-onset AD according to the criteria established by the Diagnostic and Statistical Manual of Mental Disorders (4th edition, DSM IV) [American Psychiatric Association, Diagnostic and Statistical Manual of Mental Disorders, American Psychiatric Association, Washington, DC, USA, 4th edition, 1994], the National Institute of Neurological and Communicative Disorders and Stroke, and reevaluated according to the NIA-Alzheimer’s Association workgroups on diagnostic guidelines for AD [G.M. McKhann,D.S. Knopman,H. Chertkowetal.,“The diagnosis of dementia due to Alzheimer’s disease: recommendations from the National Institute on Aging-Alzheimer’s Association workgroups on diagnostic guidelines for Alzheimer’s disease,” Alzheimer’s and Dementia, vol. 7, no. 3, pp. 263–269, 2011].

In the revised ms we report (see pag 4, lines 88-93): Recent studies report that causative gene mutations associated to familial AD, have been identified in APP, PSEN1 and PSEN2. However, mutations in these genes are able to explain just a small percentage of all fAD cases. This finding suggests the existence of other, inherited, disease-predisposing genes (Cacace, R.; Sleegers, K.; Van Broeckhoven, C. Molecular genetics of early-onset Alzheimer’s disease revisited. Alzheimers Dement. 2016, 12, 733–748). Thus, since we did not find any mutation in the main candidate genes (APP, PSENs), further experimentations will be needed to search for gene/s possibly involved in fAD1 phenotype.

  1. The authors described as “we found that the mature/immature APP ratio was significantly lower in both fAD fibroblast cell lines” in page 5 line 110-111. This result (with statistical analysis?) should be reflected in the Figure 1.

Response: We agree with the referee’s comment. The APP ratio analysis has been now reported in a new Table2 in the revised version (pag 6, line 128).

  1. The authors used Transferrin-conjugated Alexa 594 and secondary antibodies conjugated with Alexa-488 and Alexa-546 for immunostaining according to the description in the method section. I have doubts about the possible crosstalk artifacts since there is overlap in the spectra between Alexa-488 and Alexa-546 or between Alexa-546 and Alexa-594. In the Figure2, co-immunostaining of these three dyes was seemed to be performed since cell shape looks same. Have you checked your microscope setup carefully to be sure that there is no possibility for crosstalk? Otherwise, I recommend you to perform same immunocytochemical experiment (Figure3 as well as Figure2) using another combination of fluorescence-conjugated antibodies.

Response: We apologize for the concise description we made for Figure 2 in the previous text version. We thank the referee for this observation, in fact we realize that it is confusing to the readers. APP/Tfr/Giantin were labeled by a triple immunostaining using different secondary antibodies (cy2/Alexa594/cy5, respectively). The colour channels have been splitted in green, red and blue. The blue channel attributed to the third fluorophore (cy5), for an easier visualization of the overlay channel, has been rendered red. Figure legend (pag 7) and methods section (pag 18, line 428-434) have been replaced accordingly.

  1. The authors described that all statistical analyses were performed using Student’s t-test. However, Student’s t-test cannot be used for a comparison of more than two groups. The authors should perform statistical analysis using the test other than Student’s t-test.

Response: we thank the referee for the suggestion to use an alternative test for assessing statistical changes after treatment of cells with inhibitor. Accordingly, we have now replaced the statistics with 2-way ANOVA throughout the manuscript and methods section.

Round 2

Reviewer 3 Report

 I have reviewed the manuscript entitled “Inhibition of 37/67kDa laminin-1 receptor restores APP maturation and reduces Amyloid- in human skin fibroblasts from familial Alzheimer’s disease” written by Bhattacharya A et al.

 In my view, the manuscript has been revised well. I think this manuscript is now suitable for publication in Journal of Personalized Medicine.